# Double- or Triple-Tiered Protection: Prospects for the Sustainable Application of Copper-Based Antimicrobial Compounds for Another Fourteen Decades

**DOI:** 10.3390/ijms241310893

**Published:** 2023-06-30

**Authors:** Yue Yu, Haifeng Liu, Haoran Xia, Zhaohui Chu

**Affiliations:** 1State Key Laboratory of Hybrid Rice, Hubei Hongshan Laboratory, College of Life Sciences, Wuhan University, Wuhan 430072, China; 2State Key Laboratory of Crop Biology, College of Agronomy, Shandong Agricultural University, Tai’an 271018, China

**Keywords:** copper ions, antimicrobial compound, elicitor, defense response, heavy metal, plant immunity

## Abstract

Copper (Cu)-based antimicrobial compounds (CBACs) have been widely used to control phytopathogens for nearly fourteen decades. Since the first commercialized Bordeaux mixture was introduced, CBACs have been gradually developed from highly to slightly soluble reagents and from inorganic to synthetic organic, with nanomaterials being a recent development. Traditionally, slightly soluble CBACs form a physical film on the surface of plant tissues, separating the micro-organisms from the host, then release divalent or monovalent copper ions (Cu^2+^ or Cu^+^) to construct a secondary layer of protection which inhibits the growth of pathogens. Recent progress has demonstrated that the release of a low concentration of Cu^2+^ may elicit immune responses in plants. This supports a triple-tiered protection role of CBACs: break contact, inhibit microorganisms, and stimulate host immunity. This spatial defense system, which is integrated both inside and outside the plant cell, provides long-lasting and broad-spectrum protection, even against emergent copper-resistant strains. Here, we review recent findings and highlight the perspectives underlying mitigation strategies for the sustainable utilization of CBACs.

## 1. Introduction

Agriculture is the most important basic industry globally, as it plays an important role in supporting the demand for food and raw materials in other industries. However, with the global population projected to reach 9.7 billion by 2050, the demand for food by humans has also been increasing [1,2]. However, annual increases in production yield for the vast majority of crops have been steadily declining [3,4,5]. In order to meet the growing needs of humanity and industry, certain measures must be urgently taken to increase food production. Although breeding high-yield varieties is an effective strategy, the yield losses caused by crop diseases have been estimated as 11–30% [6], and the losses of fruits, vegetables, and grains caused by pests and diseases may reach as high as 78%, 54%, and 32% without fungicides [7]. Therefore, finding effective ways to prevent and control plant disease is expected to be useful in improving food production.

Copper ions are used as broad-spectrum protectant fungicides in agricultural systems to control a series of plant diseases. They appear in some forms of copper-based antimicrobial compounds (CBACs), which have been commercially used for nearly 14 decades [8]. With broad-spectrum antimicrobial activity, CBACs can control a wide range of plant diseases, such as grape downy mildew [9], citrus black spot [10], fire blight of pome fruits [11], walnut blight [12,13], potato late blight [14], stone fruit canker [15], coffee berry disease [16], olive leaf spot [17], and powdery mildew of many other crops [18,19,20]. At present, not considering metal contaminants, CBACs are still at the forefront as the main pesticides sold in Europe [8,21].

In this review, we focus on recent advances regarding the function of copper ions in CBACs, emphasizing the historical development and activation of plant immunity, which helps in understanding the physics and molecular mechanisms of copper-mediated protection to promote the sustainable future use of CBACs.

## 2. Development of Copper-Based Antimicrobial Compounds

The control of plant diseases through the use of copper agents has a long history. In fact, the earliest known record of bluestone (copper sulfate) being used to kill smut spores on wheat grains was in France [22] when, in 1807, Prevost began to use bluestone to disinfect grain seeds [22]. In 1838, the Boucherie of France found that adding 1 part copper sulfate (CuSO_4_) to 100 parts water could effectively protect wood [23]. After that, CBACs consisting of CuSO_4_ were widely used to control grapevine downy mildew (Table 1), establishing the rudiments of CBACs.

In 1873, Dreisch added a lime water bath after the application of bluestone, thus improving Prevost’s method of treating wheat grain seeds [22]. In 1883, the beneficial effect of mixing lime with CuSO_4_ was proven [24], and in 1885, the French botanist Pierre-Marie-Alexis Millardet published his famous discovery that CuSO_4_ and lime mixed could protect grapes from downy mildew [24]. This mixture became known as the Bordeaux mixture and was the first commercial fungicide made of CBACs [24]. From 1887 to 1890, extensive tests using the Bordeaux mixture were conducted at several experimental agricultural stations, and it was shown to be effective in controlling various diseases, including potato late blight and many other leaf spots and blights [24,25]. As an excellent fungicide and bactericide, the Bordeaux mixture has been widely used for the past thirteen decades all over the world (see Table 1), representing the first generation of inorganic copper fungicides.

As excess free Cu^2+^ is toxic to plants, lime can help to reduce the concentration of free Cu^2+^ and cover the surface of plant tissues more effectively and stably. The ratio between CuSO_4_ and lime has been continuously improved in the development of CBACs. In the beginning, Millardet’s 8:8:100 formula involved mixing 8 pounds of CuSO_4_, 8 pounds of hydrated lime, and 100 gallons of water; however, the concentration of free copper ions was still too high to be used on young and copper-sensitive plants. To use the Bordeaux mixture on copper-sensitive plants, the relative amount of hydrated lime was increased in the formula (to a ratio of 4:4:100) to fix the Cu^2+^. To reduce the amount of CuSO_4_ and hydrated lime, a ratio of 2:6:100 was also used for spraying copper-sensitive seedlings [26].

Since the development of the Bordeaux mixture, second- and third-generation inorganic copper fungicides have gradually taken over in the management of plant diseases, such as copper oxychloride, copper oxide, and copper hydroxide [8,27]. These inorganic copper fungicides follow the principles of progressing from high to low concentrations and from soluble to insoluble from generation to generation. To cope with the disadvantages of inorganic copper fungicides, including complex preparation processes, instability, and difficulty combining them with other fungicides, two other types—synthetic organic copper and natural organic copper—have been developed [28]. Compared with inorganic CBACs, organic copper fungicides such as Cueva copper abietate and thiodiazole–copper have low copper content and greater stability, resulting in less environmental pollution and phytotoxicity (Table 1). With technological advancement, advanced nanotechnology has been introduced into the production of CBACs. Some papers have reported that, although the concentration of copper is low in nanoparticles (NPs), they are still effective in controlling diseases in tomatoes, pepper, rice, and many other plants and have reduced impacts on the environment due to their easy uptake into plant cells [29,30,31,32,33]. Copper nanoparticles are more efficient than conventional CBACs in preventing fungal-induced diseases [34]. For example, relevant studies have shown that Cu–chitosan NPs exhibit higher antifungal activity, due to both the chitosan and copper ions. On one hand, the chitosan component of the NPs can induce plant-defense-related enzymes, leading to an increase in plant antifungal activity. On the other hand, fungi have a tendency to produce different levels of acids during their infection of plants. The resulting acidic pH induces the protonation of chitosan amino groups, resulting in the release of free copper ions from the chitosan nanostructures. These enter fungal cells and induce the synthesis of highly reactive hydroxyl radicals, which destroy biological molecules [35]. Conventional nanoparticle synthesis routes using chemical and physical methods, such as chemical reduction, hydrothermal, and sol–gel, methods are considered harmful to the environment, due to the use of toxic chemical products, and are also costly. In response, many aqueous extracts from plants such as *Portulaca oleracea* and *Piper nigrum* have recently been used to biosynthesize promising, safe, cheap, and eco-friendly Cu-NPs [21,36,37]. Moreover, recent studies have shown that copper nanoparticles in combination with conventional fungicides can provide an environmentally safe and sustainable resistance management strategy through reducing the use of fungicides [38].

**Table 1 ijms-24-10893-t001:** Various types of CBACs with advantages and disadvantages.

Type	Name	Active Constituent	Advantages	Disadvantages
Inorganic copper fungicides	Copper sulphate	CuSO_4_	Anti-microbial [8]	Phytotoxicity, Short-lasting [8]
	Copper oxychloride	3Cu(OH)_2_CuCl_2_	Anti-microbial, Stable [27]	Short-lasting [39]
	Copper oxide	CuO	Low toxicity, Stable, Anti-microbial [8]	Low efficiency
	Copper hydroxide	Cu(OH)_2_	Low toxicity, Stable, Anti-microbial [27]	Phytotoxicity [40]
Organic copper fungicides	Oxine–copper	C_18_H_12_CuN_2_O_2_	Low toxicity, Anti-microbial, Long-lasting [41]	Drug resistance, Environmental pollution, Phytotoxicity
	Thiodiazole–copper	C_4_H_4_N_6_S_4_Cu	Low toxicity, Stable, Anti-microbe [42]	
	Copper abietate	C_40_H_58_CuO_4_	Low toxicity, Stable, Anti-microbial	
Copper-based nanoparticles	CuS nanoparticles	Cu and S	Slow-release, Stable, Low toxicity, High-efficiency [43]	Drug resistance, Phytotoxicity [44]
	CuO nanoparticles	CuO		
	CuAlO_2_ nanoparticles	Cu and Al		

## 3. Construction of a Physical Barrier by Covering Plants with Slightly Soluble CBACs 

A long historical practice is the application of CBACs as slightly soluble protective reagents, which cover the surface of plant tissues before diseases emerge and form a film to prevent direct contact between pathogens and plants. Moreover, they are absorbed into the tissue surface, making them difficult to be washed away by rain and dew, thus maintaining a long-term residual effect. In 1882, Millardet proposed that the actual treatment of mold with a mixture of CuSO_4_ and lime should not aim to kill the parasites in the leaves but, instead, should aim to prevent their development by covering the surface of leaves with various substances [45]. In the Bordeaux mixture, the generated calcium sulfate is thought to be necessary for tightly adhering the CBACs to the leaves. In other microsoluble CBACs, specific chemical additives facilitate this attachment [8,46]. In addition to separating the host plant from pathogens, the microsoluble film has two additional benefits: On one hand, the forms of CBACs that mainly coat the plant surface are soluble but complexed, thus only allowing a few free copper ions to be released and control plant diseases [47]. In fact, the concentration of copper ions on a leaf depends on the equilibrium established with complex and soluble copper forms [48]. This prevents the release of an excessive amount of Cu^2+^, which would lead to plant phytotoxicity. On the other hand, the slow and continuous release of Cu^2+^ on the tissue surface provides long-term prevention for plants; however, this kind of protection is not stable. Long-term rainwater scouring can break through the film of the Bordeaux mixture, reducing the protective effect and efficacy and allowing invasion by pathogens. Moreover, the weak acid substances secreted by plants and micro-organisms can also generate an acidic environment, leading to the inappropriate high-frequency release of Cu^2+^, which may have a negative phytotoxic effect [49]. High concentrations of Cu^2+^ can create visible corky damage on the surface of young fruit, reducing the aesthetic value of the fruits and compromising their marketability [8].

To reduce the environmental pollution and phytotoxicity caused by excess Cu^2+^, it is particularly important to develop novel CBACs. In fact, the release rate of Cu^2+^ affects the availability and persistence of conventional CBACs, such as the Bordeaux mixture. The rapid release of Cu^2+^ can have a good effect in terms of disease management, but the pesticide effect will be short and the security poor. Due to the large particle size and water solubility, CBACs can only form discontinuous deposits on the surface of plants, allowing for only partial blocking of direct contact between pathogenic microorganisms and the plants. In contrast, thicker deposits can increase the risk of excessive release of copper ions, causing plant toxicity [8]. Therefore, developing organic copper agents to reduce the excessive release of active copper ions and/or reducing the particle size of CBACs to promote the formation of a continuous film on plant surfaces are effective strategies to prevent bacterial and fungal spore invasion. Microscopically, oxine–copper is composed of copper ions and oxine rings: two oxine rings tightly grip the copper ions, which can gradually and safely release free copper ions [41]. The small size of these particles leads to a high surface-area-to-volume ratio, meaning more uniform coverage and better protection. Additionally, smaller particles are more tightly adsorbed on the plant surface and are more tolerant to rain wash than larger particles, giving longer effective protection. Studies using SEM have shown that foliar application of MoS_2_-CuNPs allowed for the formation of a protective film and increased the density of trichomes on the surface of rice leaves, thus preventing infection by *Xanthomonas oryzae* pv. *oryzae* cells [50]. Furthermore, NiO:Cu thin films observed by SEM presented antifungal activity against *Aspergillus niger* (which affects various fruits) and *Macrophomina phaseolina* (which is a soil-borne fungus responsible for root and lower stem infections in several plants) [51]. In addition to the smaller particle size, the diversity of forms of CBACs and their additives is another method to ensure even spraying, promoting adherence and stronger fixation on plant tissues. At present, various forms of CBACs, including aqueous solutions, wettable powders, and suspending agents, are broadly utilized. These have good efficacy but often a poor retention period, being greatly affected by rain wash. Some researchers have developed mineral oil emulsions for CBACs. The addition of mineral oil to fungicide spray mixtures is a frequently used strategy for the control of citrus black spot and potato pests [52,53], as mineral oil can significantly improve the diffusion, adhesion, and retention of copper ions; increase the deposition amount of effective components; and improve the ability to resist rain wash after mixing with CBACs [54]. Overall, it should be emphasized that mineral oil can highly improve the prevention effect of CBACs.

In addition, adjuvants are the key factors for improving the stability and efficacy of CBACs. During the processing and application of pesticides, surfactants can help them to distribute over, adhere to, and penetrate the surfaces of plants, directly or indirectly improving the effective pesticide usage rate. Agricultural organosilicon adjuvants, such as Silwet L-77 or siloxane, are often used as adjuvants for CBACs for the control of citrus canker due to their good wettability, ductility, and permeability. When mixed with CBACs, adjuvants can improve the ductility and adsorption properties of copper agents on the leaves, increasing the tolerance to rain acidification and plant disease resistance [46]. Alternatively, bamboo vinegar—which contains organic acids, ketones, and alcohols—is a good solubilizer, co-solvent, and penetrant. Researchers have boiled bamboo vinegar and CuSO_4_ to form a preparation which can enhance the control effect of CuSO_4_ on tobacco brown spot disease and black shank disease, as well as improving its inhibition of the growth of green algae, while the copper ion concentration remains unchanged [55]. Additionally, ethoxy-modified polysiloxane, polyoxyethylene monolau-rate β pine terpene polymer, ammonium salt, and other adjuvants play supporting roles to CBACs, helping them to attach to plant tissues more evenly and stably [46]. However, recent studies have also revealed that Cu^2+^ released from CuSO_4_ and nanomaterials is rapidly absorbed into the leaf cuticle [56]. Interestingly, copper-based nanoparticles can pass quickly through the cuticle, while CuSO_4_ can stay longer in the leaf cuticle, which appears to strengthen the alternative physical barrier [56].

## 4. The Second Tier of Protection and Copper-Resistant Strains

Generally, copper is a necessary metal ion for bacterial growth and development, and bacteria uptake Cu^2+^ into the cytoplasm through copper uptake transporters such as CcoA and YcnJ-like proteins [57,58,59,60]. Then, the copper reductase cbb3-type cytochrome c oxidase (cbb3-Cox) assembly factor CcoG is present on the cell membrane, where the incoming Cu^2+^ is assembled into the cysteine conservative motif of CcoG and converted into Cu^+^ through transferring electrons to the [4Fe-4S] cluster (Figure 1a). Furthermore, Cu^+^ binds to the active center of enzymes to maintain its vital role in bacteria [61,62]. However, excess free copper ions also lead to toxic or antimicrobial activity in bacteria, forming the second-tier protection of CBACs. The antimicrobial activity of CBACs can be further divided into two parts: first, deposited CBACs can react with water and oxygen to produce OH^−^, causing bacterial cell membranes to suffer from oxidative damage, leading to protein denaturation and increased membrane permeability. This damage to cell membranes further results in the leaking out of some bacterial essential nutrients and proteins. In addition, excessive Cu^2+^ or Cu^+^ entering the cytoplasm will cause bacterial oxidative stress and even cell death. Under an anoxic environment, Cu^+^ replaces iron in the iron–sulfur clusters of dehydratases, resulting in the degradation of those crucial enzymes. Furthermore, the released iron may subsequently initiate the Fenton/Haber–Weiss reaction, while the transformation between Cu^+^ and Cu^2+^ leads to a Fenton-like reaction, all of which generate OH^−^ and ROS, consequently causing lipid peroxidation, protein oxidation, and nucleic acid damage [63,64,65,66]. 

Bacteria also depend on two systems to overcome excess copper: copper homeostasis and copper resistance protein (*cop*). As shown in Figure 1, in order to maintain the cytoplasmic copper concentration, bacteria have developed three strategies: First (I), the chaperone protein CopZ loads Cu^+^ and transfers it to P1B-type Cu-exporting ATPase CcoI and P-type ATPase family CopA, which respond to the efflux of excessive Cu^+^ [58,67,68,69,70,71]. Second (II), metallothionein (MT) is a super-family of cysteine-rich small proteins which bind Cu^+^ (as well as other heavy metal ions) through metal–sulfur bonds in order to neutralize their toxicity [72,73,74,75]. Third (III), excess Cu^+^ is bound with the elevated cytosolic copper storage protein (Ccsp), which consists of a homotetramer assembly capable of binding Cu^+^ with the help of a CopZ-like copper chaperone [76].

Over-use of CBACs has resulted in long-term exposure of plant pathogenic bacteria to high concentrations of copper ions, resulting in the selection of copper-resistant strains, such as *Xanthomonas, Pseudomonas,* and *Erwinia* spp. [77,78,79,80,81,82,83], which have direct and indirect impacts on agricultural production [84,85]. Through the isolation and identification of copper-resistant strains from copper-rich soil, more than 95% of the copper-resistant isolates were identified as Gram-negative bacteria [86]. Unlike Gram-positive bacteria, Gram-negative bacteria have an outer membrane, a periplasmic space, and an inner membrane which endow them with a special structural basis for copper resistance. To date, most copper-resistant strains have developed from horizontal transfer of the *cop* system or *pco* system in response to excessive copper, with representative examples being *Pseudomonas syringae* pv. *tomato* (*Pst*) and *Escherichia coli*, respectively [87].

The cop system is a conserved copper-resistant system in *P. syringae* pv. *tomato,* which is encoded by an operon containing up to six genes (*copABCDRS*) on the plasmid pPT23D [87]. CopA is a periplasmic protein that contains methionine, histidine, and aspartic-acid-rich motifs. Each CopA protein can combine up to eleven copper ions [88,89]. This high binding capacity restricts excess copper from entering the bacterial cytoplasm. As an outer membrane protein, CopB also contains repetitive amino acid sequences (Asp-His-X2-Met-X2-Met). Although there is no direct evidence that CopB can combine with copper [90,91,92], it may be assumed that the role of CopB is to fix extracellular copper ions. CopC is a periplasmic chaperone protein that contains two copper ion binding sites for binding either Cu^+^ or Cu^2+^. It has been proposed that CopC transfers Cu^+^ to different interactors, such as CopA, CopB, CopD, and CopS, in order to balance the Cu^+^ concentration in cells [92]. On one hand, CopD, the interaction protein of CopC, is a plasma membrane protein that transports essential CopC-delivered copper through the inner membrane into the cytoplasm [89,90]. On the other hand, CopC interacts with CopA and CopB to deliver the carried Cu^+^ for fixing, which can reduce the associated toxicity [90]. The CopS located on the plasma membrane serves as a copper sensor, which may interact with CopA or CopC to transmit copper signals to CopR, thus continuously regulating the expression of the *cop* operon activated by CopR [91]. To summarize the copper resistance mechanism of *Pst*, the *cop* operon located on plasmid pPT23D can chelate excessive copper ions through a group of proteins, particularly CopA and CopB. The copper resistance mechanism in *E. coli* is the *pco* system containing *pcoABCD*, which is located on the plasmid pRJ1004 and corresponds to the *cop* operon in *Pst* [93,94,95,96]. To date, all identified copper-resistant *Pseudomonas* strains have homologs of the *cop* operon in their chromosomes [97]; for example, *Cupriavidus metallidurans* CH34 contains the complete *copABCDRS* [98,99], while *X. citri* pv. *citri* contains only *copABCD* [100]. Some Xanthomonas copper-resistant strains only contain *copLAB*, conferring resistance to copper ions [101,102,103]. The mechanisms of copper-resistant fungi have been reviewed in [104], which were generally indicated to enhance Cu^+^ exporting and homeostasis. *Yarrowia lipolytica* is an inherently copper-resistant yeast in which Cu^2+^ significantly promotes the yeast-to-hypha transition, allowing for the better survival of hyphae than yeast-form cells in the presence of CuSO_4_ [105].

Some pathogens could alternatively develop new weapons to overcome their copper sensitivity. *Xanthomonas oryzae* PXO99A is more sensitive to copper than other wild strains caused by the *copA* mutation [106]. It seems not to back mutate for a copper-resistant strategy but to develop a novel TAL effector of PthXo1 to upregulate the expression of *OsSWEET11/Xa13* in rice [107], which interacts with plant copper-uptake complex components of OsCOPT1 and OsCOPT5 to reduce the copper concentration in vascular tissue [108]. Moreover, the upregulated OsSWEET11 protein has additional susceptible functions as the sucrose efflux from the phloem parenchyma cells for bacterial proliferation [109]. Consistent with the conclusion, PXO99A introduced into the *copAB* could restore copper resistance but fail to overcome the *xa13*-mediated resistance [106]. 

**Figure 1 ijms-24-10893-f001:**
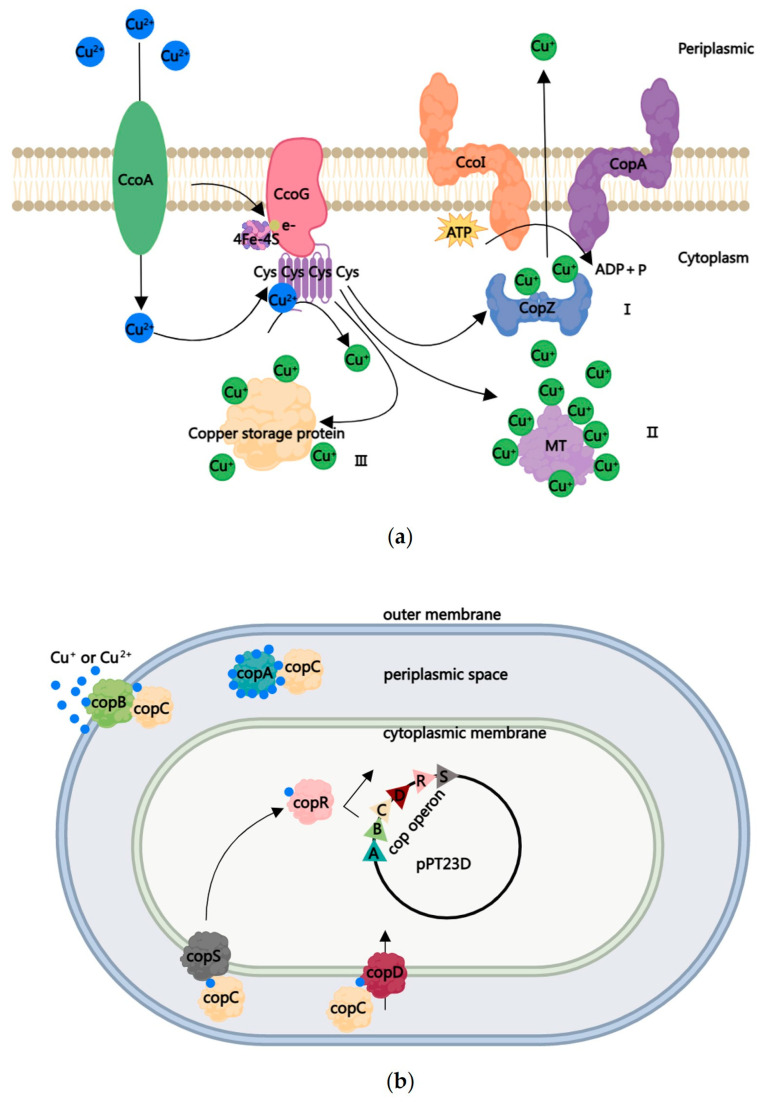
Two strategies used to manipulate the excessive copper in Gram-negative bacteria. (**a**) Three strategies for copper homeostasis: (I) the chaperone protein CopZ loads Cu^+^ and transfers it to P1B-type Cu-exporting ATPase CcoI and P-type ATPase family CopA, which respond to the efflux of excessive Cu^+^; (II) each metallothionein (MT) protein binds seven Cu^+^ ions to neutralize the toxicity; and (III) copper storage protein binds excessive Cu^+^. (**b**) Cop systems to resist copper. *P. syringae* pv. *tomato* strains encode an operon of *copABCDRS* containing up to six genes on the plasmid pPT23D. The periplasmic protein CopA combines eleven copper ions to restrict the excess copper in the cytoplasm. The outer membrane protein copB may play a role in fixing extracellular copper ions, as does CopA. The periplasmic chaperone protein CopC contains two copper ion binding sites, for binding either Cu^+^ or Cu^2+^. CopC delivers Cu^+^ to different interactors—such as CopA, CopB, CopD, and CopS—either for uptake or to fix Cu^+^ to balance the concentration in cells. The plasma membrane protein CopS acts as a copper sensor. It may interact with CopA or CopC to transmit copper signals to CopR, thus continuously regulating the expression of *cop* operon activated by copR (modified by references [87] and [91]). The picture was drawn using the MedPeer software (https://user.medpeer.cn/ (2 February 2023)).

## 5. Enhanced Plant Resistance to Pathogens and Activation of Defense-like Responses

At present, the application of copper preparations is the only way to control many plant diseases. However, an over-reliance on and over-use of CBACs have resulted in the evolution of many Cu-resistant strains, reducing their effectiveness in controlling plant diseases [110,111]. Developing new types of CBACs or mixing them with other fungicides provides an effective way to control plant diseases caused by copper-resistant strains. Copper used with EBDCs (Ethylene bis-dithiocarbamates) can provide control of bacterial speck and spot diseases, even when copper-tolerant populations are present [112,113,114]. Recent studies have demonstrated that advanced copper composites and nano-magnesium oxide materials are effective against copper-tolerant *Xanthomonas* spp., increasing the control of bacterial spot in tomato under field conditions [115,116,117]. Furthermore, although the emergence of copper-resistant strains has rapidly increased, CBACs are still effective in controlling some of the diseases caused by those pathogens [118], implying that the protection imparted by CBACs does not only rely on their antimicrobial activity and the formation of a physical film. Previous reports have demonstrated that copper stress could also activate a series of defense-like responses in plants. In alfalfa (*Medicago sativa*), the mitogen-activated protein kinases (MAPKs) SIMK and SAMK are involved in the response to pathogen-associated stimulation. Excessive copper specifically activates the MAPK SIMKK, which can activate SIMK and SAMK in *Medicago sativa* [119]. Additionally, excess copper activated MAPK2, MAPK3, and MAPK4 in rice roots [120,121,122]. ROS burst is one of the earliest events in plants under heavy metal stress, as the ROS act as signaling molecules that regulate the plant’s response to abiotic and biotic stresses [123,124,125]. Excess copper promotes ROS synthesis through Fenton and Haber–Weiss reactions [126]. Copper-elevated ROS accumulation has already been identified in *Arabidopsis thaliana* [127], *Pisum sativum* L. [128], *Medicago sativa* [129], and *Oryza sativa* L. [130]. Previous reports have demonstrated that copper stress enhanced plant resistance to pathogens through copper-binding proteins. In cotton, the blue copper-binding protein GhUMC1 has been shown to be involved in resistance to *Verticillium dahlia* through regulating the jasmonic acid signaling pathway and lignin metabolism [131]. In barley, *Mla* and *Rom1* negatively regulate miR398, which elevates the transcription level of *SOD1* and enhances resistance against powdery mildew [132], indicating the important role of the miR398–SOD module in regulating plant resistance against pathogens. Interestingly, the foliar application of two copper nanomaterials enhanced resistance to *Fusarium oxysporum* f. sp. *lycopersici*, a pathogen that causes the root fungal disease Fusarium wilt, as well as enhancing phenylalanine ammonia-lyase (PAL) and peroxidase (POD) activities in tomato roots [30,56].

## 6. Eliciting Plant Immunity to Strengthen the Third-Tier Barrier

The above observations—that is, that CBACs can manage copper-resistant strains and excessive copper can trigger defense-like responses in addition to being toxic to plants [118,119,120,121,122,123,124,125,126,127,128,129,130]—indicate that copper may directly trigger plant immunity. Indeed, Liu et al. found that a concentration of 10 nM CuSO_4_ was sufficient to enhance the resistance of *Arabidopsis* plants against *Pst* DC3000 [133]. In addition, spraying potato with CuSO_4_ (100 nM) enhanced resistance to late blight [134]; however, in in vitro co-culture experiments, these concentrations of CuSO_4_ had no inhibitory effect on microbial growth [133,134]. Moreover, they found that copper ions triggered a series of immune responses, including ethylene (ET) and salicylic acid (SA) biosynthesis pathways, ROS burst, Ca^2+^ signaling, MAPK activation, callose deposition, and up-regulation of the expression of pathogenesis-related (*PR*) genes [133], which are similar to the responses induced by flg22, a conserved short peptide of flagellin from *Pst* DC3000 [135,136]. In contrast with flg22-triggered immunity, Cu^2+^ treatment rapidly activated the synthesis of ET by specifically inducing the expression of *AtACS8* dependent on the CuRE *cis*-element in the promoter region [137]. Downstream of the ET signaling pathway, Cu^2+^-mediated callose deposition required both *AtMYB51* and *AtMYB122*, while it mainly required *AtMYB51* for flg22 in *Arabidopsis* [138]. A nuclear copper chaperone CCP containing the classical copper-binding site may interact with and recruit the transcription factor TGA2 to induce the expression of *PR1* and enhance the resistance to *Pst* DC3000 [139]. In potato (*Solanum tuberosum*), Cu^2+^ activated ET biosynthesis to induce resistance to potato late blight, as well as inhibiting the biosynthesis of abscisic acid (ABA) by activating the transcription factor *StEIN3* (ethylene insensitive 3), thus directly repressing the expression of *StNCED1* (9-cis-epoxycarotenoid dioxygenase) and the ABA biosynthesis gene *StABA1* by targeting their promoters [134]. Yao et al. have recently found that copper ion transporters and copper ion binding proteins, such as HMA5, were significantly induced and played a broad-spectrum role in virus–rice interactions. Most of the copper ions entered rice cells from the intercellular space, increasing the copper ion content in the leaves. Copper-orchestrated virus resistance was promoted through inhibiting the accumulation of the SPL9 protein, thus reducing the expression of SPL9 target gene miR528 and enhancing the transcription level of *ascorbate oxidase* (*AO*) and ROS levels [140]. On the other hand, copper ions could directly activate the AO enzyme activity to enhance viral resistance in rice [140]. Without a doubt, a low concentration of copper ions can trigger plant immune responses, thus participating in the construction of a third-tier barrier to protect plants against pathogens. Similarly, the induction of plant immunity was observed when using Cu_2_O-NPs to control cucumber root rot disease [141]. However, over thirteen decades, a considerable number of studies have shown that CBACs cannot effectively control the plant diseases caused by copper-resistant strains compared with copper-sensitive strains [142,143]. Inappropriate timing of applications, with respect to wounding and infection events, is an alternative explanation; that the activated PTI-like immunity may not be able to control all pathogens is also an alternative explanation. However, such results suggest that more research is needed to fully explain the specific mechanisms by which copper ions regulate plant immune responses, as well as the need for further research on whether copper can trigger immune responses in different kinds of plants.

## 7. Summary and Future Prospects

As a metal ion, copper is the main component of commercial CBACs. At present, the mechanisms of CBACs can be summarized into two- or three-tiered protection (Figure 2), detailed as follows. First, the slightly soluble CBACs form a dense protective film on the plant surface, which acts as a physical barrier to prevent contact between the invasive pathogenic microorganisms and the host; second, the released ionic copper destroys the cell membrane of the pathogenic bacteria, leading to the leakage of nutrients, denaturation of various proteins, and inactivation of enzymes, thereby killing the microorganisms; and third, copper ions can also stimulate plant immune responses to further strengthen the immunity of the host plant. Such a three-tiered protection provides a perfect design for broad spatial disease resistance, supporting the application of CBACs for more than thirteen decades.

However, with increases in rain acidification and copper-resistant strains, it is necessary to constantly innovate relevant methods and technologies in order to optimize the application of CBACs. Based on our knowledge, prospective studies can be carried out at the following three levels. First, combinations of systemic fungicides and/or plant stimulants need to be broadly investigated. For example, thiodiazole mixed with copper to produce commercial thiodiazole–copper can recover the pathogen inhibition activity while reducing the usage of ionic copper [42]. Various plant stimulants have been widely used in agricultural production, some of which have been shown to possess novel bioactivities [49,144]; however, they normally have lower efficacy in reducing disease incidence and severity compared to CBACs [144]. Therefore, the combination of such stimulants with CBACs is worth investigating in future research.

Second, the long-term and excessive use of CBACs has caused the deposition of copper in the soil and environmental pollution [21,144]. Therefore, in order to ensure the safe use of CBACs for another 13 decades (or even longer), it is imperative to further reduce their usage, together with their tolerance to scouring by rain. As an advanced fungicide, Cu-NPs have smaller particle size and higher surface-area-to-volume ratio, and they can pass more quickly through the cuticle than traditional CBACs [56]. Therefore, Cu-NPs have attracted extensive attention in agricultural applications. Scientific researchers have revealed the positive effective roles of Cu-NPs in controlling diseases, reducing toxicity, promoting growth, and increasing ion content in rice seeds [29,30,31,32,145]. In addition, with technological improvement and development, the cost of Cu-NPs and plant-based CuO-NPs can be expected to gradually decrease [21,36,145], laying the foundation for the long-term use of CBACs. Along with the innovation of advanced production technologies, traditional CBACs are improving in a more stable, low-toxicity, and environmentally friendly manner. Scientists may develop novel adjuvants to increase the ductility, adhesion, and permeability of CBACs, allowing for a reduction in the content of ionic copper in the CBACs. In general, reducing the cost of Cu-NPs and novel additives may help CBACs to achieve better development and applications in the future.

Finally, although copper-triggered plant immunity has been reported, the signal transduction pathway(s) associated with such induced resistance remains unclear. Further detailed studies on the specific mechanisms underlying copper-triggered plant immunity should be conducted in order to better utilize CBACs, including the development of pesticide application techniques and the cultivation of ideal crop varieties that are more rapidly and strongly responsive to ionic copper than current versions. In general, once the above three problems are effectively solved, CBACs can be expected to serve humanity’s agricultural purposes for another thirteen decades.

## Figures and Tables

**Figure 2 ijms-24-10893-f002:**
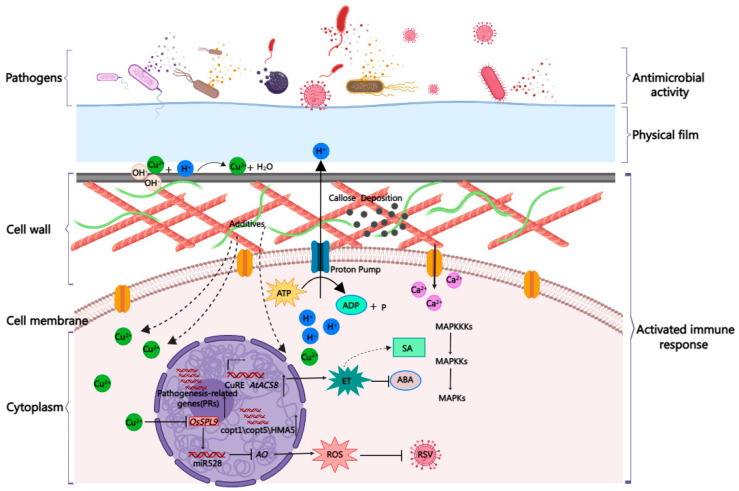
Illustration of the three tiers of protection provided by copper-based antimicrobial compounds. The spatial defense network consists of a physical film, antimicrobial activity, and activation of immune responses. ABA, abscisic acid; AO, ascorbate oxidase; ET, ethylene; ROS, reactive oxygen species; SA, salicylic acid. The picture was drawn using the MedPeer software (https://user.medpeer.cn/ (6 February 2023)).

## Data Availability

Not applicable.

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
