# Peer review of "Double- or Triple-Tiered Protection: Prospects for the Sustainable Application of Copper-Based Antimicrobial Compounds for Another Fourteen Decades"

_ijms, 2023, doi:10.3390/ijms241310893_

Round 1

Reviewer 1 Report

This review has addressed the mechanisms of action and issues involved in the use of copper containing compounds for the control of plant pathogens. The authors are correct in that copper containing compounds remain one of the few effective bactericides for use on most crop plants, although they should note that they are notoriously poor in their efficacy for the control of plant pathogenic bacteria. The world has long wanted to have much better copper bactericides, and so far, the modifications of copper containing compounds, including copper Nanoparticles have not been the panacea that this review would seem to suggest that they are. The review contains a lot of information, but is very awkwardly written. The review is in desperate need of grammatical improvement. There are literally hundreds of places where the text is either confusing or inaccurate the way it is written. There is no way that I can point out these many many places where improvements are needed. More importantly, I am concerned with the overall organization of story about the use of copper on plants – especially their claim that there are basically three tiers in which copper interacts with the microbes in the plant to prevent disease. I do not agree with this 3-tier concept, and do not see any differentiation between what they call tier 1 and tear 2, where they feel that some sort of physical film forms the first barrier to entry into the plant followed by the direct antimicrobial activity of the copper ions. I don't think there is any evidence that a physical barrier is being formed by the application of these copper based compounds. With the possible exception of some of the organic copper compounds, nearly all copper-based compounds are insoluble salts, such as copper oxide and copper hydroxide, or even Bordeaux mixture which is deposited as particles on the plant surface. It is hard to believe, and there was no evidence presented, that these compound present a complete physical shield on the plant surface, and instead are probably discontinuous deposits of these compounds which, by slow dissolution, release toxic copper ions onto the plant where they interact with the pathogen. In that regard, I was very disappointed that the authors did not report some of the older, what I might consider foundational work, on the chemistry of copper on leaves, such as that shown in a couple of older manuscripts by Menkissoglu. This work demonstrated that deposits of compound such as copper hydroxide left relatively large amounts of the insoluble salt on the leaf surface while there is only vanishingly small concentrations of soluble copper ions which themselves are sufficient to account for the toxicity even at such low concentrations. It also documented relatively persistent deposits of these insoluble materials on the plant. Here and there through the manuscript, the authors did come around to talking about these reservoirs of insoluble compounds which then release the active ions, but this seem to have been lost amongst a lot of other discussion. Nowhere did I see any discussion of the evidence from the early work which revealed the vanishingly small concentrations of copper ions that are sufficient to kill microbes in relationship to any of these other issues that the authors have brought up. The discussions of copper homeostasis in both bacteria and in plants was extensively covered, but it seemed largely to not have much relevance to the discussion of either the toxicity of copper ions to target pathogens, nor of copper as an inducer of resistance mechanisms within plants. Perhaps the most novel aspect of the review was the discussion of what they call the third tier of control, namely activated immune responses in the plant by the presence of copper. This was an aspect of copper that I was not fully aware of, and might be of interest to readers. However I feel that the importance of this mechanism may have been somewhat over emphasized in this review. For example, there are numerous literature reports, most of which were not excited here, which have shown that the application of carbon containing compounds such as copper hydroxide to plants that are inoculated with copper resistant strains of either Xanthomonas or Pseudomonas, did not lead to measurable control of those pathogens. Thus, if copper was actually inducing resistance mechanisms in the plant, one would have expected that some level of disease control would have been achieved by the application of copper compounds to plants irrespective of whether the pathogen itself was susceptible to the copper. This has been shown in many plants such as peppers, beans, walnuts, and a variety of other crops - namely that copper resistance strains are not influenced by copper applications. The author need to address this information  to give a balanced overview of the copper story. Another aspect of the review that I thought was not well developed, was the extensive information which reveals that the efficacy of copper compounds to control copper resistant organisms can be dramatically enhanced by mixing the copper compounds with various other pesticides such as ethylene bis dithiocarbamate fungicides such as Mancozeb etc. There is an extensive literature that indicates that copper compounds mixed with EBDC compounds improves the efficacy of controlling not only copper resistance strains, but also highly increases the efficacy of the control of copper sensitive strains that are otherwise controlled, although poorly, with copper compounds alone. On line 338 the authors mentioned the use of t hiodiazole-copper mixtures to improve control, but do not even provide a reference for this. I also felt that there was a rather superficial coverage of the use of copper nanoparticles as pesticides. Perhaps there is not enough information to provide a thorough review of these new formulations of copper, but I think it would be important for the authors to address whether the nanoparticles are simply another step down in the formulation of insoluble copper compounds by further reducing their size sufficiently so that they are better adhered to leaf surfaces, and provide more surface area from which the dissolution of the insoluble copper could occur, or, as I might expect, nanoparticles, because of their tiny size, have a different manner in which they can influence pathogens. That is, can the nanoparticles directly penetrate the bacteria and therefore deliver the copper into the cell, or are they toxic for another reason independent of the presence of dissolved copper ions. Likewise, when the authors have suggested that the copper nanoparticles actually introduced copper into the plant - what form is this copper present? They note on lines 312 to 313, that copper nanoparticles seem to induce plant immunity but was this really due to the presence of soluble copper ions that were released after introduction or due to some unique biophysical effect of the nanoparticles themselves? As noted above, the manuscript is in need of extensive revision to improve the grammar as well as to address the issues raised above in terms of organization and lack of breadth. While I did not read reference number 23 (senior author of Jones), which is cited here, but this other recent review of the use of copper compounds for disease control, (which has a title almost exactly the same title as this newreview by the way, especially as it refers to the “13 decades of use”), likely already covers many of the issues raised in this review. 

Specifics

I will attach a .PDF of a copy of my scribbles that I made on a hard copy of this manuscript. There are way too many places where I found issues that needed to be clarified or better explained, and I will attempt to discuss some of them below, but it will include many other underlined or circled text that is either an accurate, needing further explanation, etc.

Line 11. A frequent reference to “13 decades” I found to be awkward and it also has apparently been copied from the title used in a similar review published in 2018.

Lines 12 through 13. It is very awkward to refer to “high solubility to scarcely dissolvable reagents”

Line 18 “...break the contacts” is very awkward.

Lines 25 through 40 I thought it was awkward to start a discussion of copper-based pesticides with a discussion of copper homeostasis and copper biochemistry in plants. If this is needed at all, it should be introduced later. I didn't really see much of a need for this in this review, perhaps it would fit best only in the very last section about copper-induced resistance to pathogens.

Line 93. I have no idea what they mean when they refer to “plant synthesized CuO-NPS”. I have a hard time believing that plants were somehow directed to make such nanoparticles. It's definitely needs more explanation/clarity.

Lines 107 through 116. This would be a good place to refer to older literature by Menkisoglu and others which have talked about the chemistry of copper on plants. I think the authors also need to better distinguish between leaf surface concentrations of copper, which would not necessarily directly interact with living plant cells because of the barrier of the cuticle, and copper that would enter within the plant. Later they talk about various surfactants which would introduce copper into the plant. I would be very surprised whether many plants could tolerate the presence of even quite insoluble copper compounds if it was introduced into the interior of the plant.

Lines 117 through 120. This sentence seems to be out of context in that copper compounds are normally applied only to the foliage of plants to inhibit foliar pathogens, and I do not see any relevance about how copper would affect plant roots, unless they are talking about excess copper ions in the soil. This needs to be clarified.

Lines 133 through 136. It is very awkward to refer to “even spraying” (perhaps they mean uniform coverage of the leaves) and I also have no idea what an "opacifying agent" is.

Lines 137-142. I'm unfamiliar with the mixing of oils and copper-based pesticides for the control of plant disease. There do not seem to be any references that they cite here. This needs to be better documented.

Lines 143 through 149. I am very surprised at their discussion of the use of organo- silicone surfactants having relatively low surface tension could be used in conjunction with copper based pesticides. I would expect that this strategy would be unacceptable on many plants that are susceptible to copper ions. The lone reference that they give for the use of organo-silicone surfactants and coppers was for use on citrus, and I would not have expected very much penetration of the leaf with the organo-silicon surfactnts on this species because of the thick cuticle of a typical citrus plant, so I think it may be dangerous to be extrapolating the efficacy of this strategy to other crop species. In fact, on citrus itself, the better dispersal of the applied copper compounds to the very waxy leaves may well have been the main mechanism associated with the increased efficacy when combined with the surfactant.

Lines 181 through 184. I would have thought that such a sentence would be one of the very first ones used in the review, rather than buried here most of the way through the manuscript.

line 186 through 198. There is something wrong with this paragraph. In the first couple of sentences they talk about the effects of copper on pathogenic microbes, but on lines 197, they talk about plant responses. This paragraph needs to be reorganized.

Lines 206 through 209. I think the authors have focused too extensively on a single mechanism of resistance, and I don't think that exclusion of copper is the main mechanism of the resistance in many copper resistant bacteria, and in fact, it was my impression that the Pt23 system that they focus a lot on, actually confers resistance primarily by the sequestration of copper after entry. I think they need to go back and better distinguish the different mechanisms of copper resistance  - by copper export pumps that prevent accumulation, and that of copper chelating mechanisms that reduce its toxicity after entry into the cell.

Line 327 through 331. Since I really don't believe in there are three tiered mechanism or resistance, I think Figure 2 needs to be altered to reflect the antimicrobial activity which seems to be main process operating before pathogens enter the plant.

Lines 347 through 349. As noted above, I think the authors need to give us more of a background on what is known and what is not known about how copper nanoparticles work. That is, hopefully there is more information than it is currently cited about how efficiently they enter plants and in what form they enter plants etc and whether their toxicity is due to copper ions or special features of the small physical particles themselves.

Author Response

Dear reviewer,

Special thanks to you for your good comments, that is really guided us to read and think more. I have point-to point responded to each of you comments as following. And the manuscript have been asked for professional language editing as your suggestion.

  1. Response to comment: The triple-tired protection of CBACs may be a debatable point, and there hasn’t any evidence that a physical barrier is being formed by the application of these CBACs.

Response: Thank you for your question. We have changed the name to “Double- or triple-tiered protection: prospects for the sustainable application of copper-based antimicrobial compounds for another fourteen decades” in the revision version. We feel very sorry for not providing a detailed explanation of the first layer of protection. First, not all CBACs need to deposit on tissue surface, only those slightly soluble CBACs such as Bordeaux, Kocide often deposit on tissue surface to form a physical barrier. As protective fungicides, CBACs act as a protective barrier and prevents infection from occurring, which should be sprayed on plants before pathogens arrive or begin to develop. Therefore, both barrier and sterilization are necessary for CBACs. Many previous reports integrated the first and the second layer together. We discuss them separately, which does not mean that each layer exists independently. On the contrary, we believe that each part needs to cooperate with each other to make CBACs more effectively. Normally, not every CBACs has such a three-layer barrier at the same time. Therefore, in order to better develop CBACs, the three-layer mechanism should be considered as much as possible. Up to now, the development of CBACs has gone through three stages: inorganic fungicides, organic fungicides, and nano fungicides. Studies have indicated that Cu-NPs can evenly cover the surface of plants and separate pathogenic bacteria by using SEM, which I have listed in this article. All issues help to support our viewpoint of a double- or triple-layer protection mechanism.

  1. Response to comment: This manuscript didn’t report some older manuscripts by Menkissoglu

Response: We are very sorry for the negligence in not listing this part of the work, and we have already added it to the manuscript.

  1. Response to comment: This manuscript may over emphasized plant immunity induced by CBACs. There are numerous literature reports have shown that the application of carbon containing compounds such as copper hydroxide to plants that are inoculated with copper resistant strains of either Xanthomonasor Pseudomonas, did not lead to measurable control of those pathogens.

Response: Thank you for your suggestion. For CBACs, two- or three layer of protection is an integrated mechanism. Each one is very critical and important, as long as any layer can interrupt the occurrence of plant disease. First, the recent studies of Liu et al and Yao et al, which we have listed in this manuscript, provided strong evidence that copper ions can induce plant immune responses. We don’t over emphasized the copper-induced plant immunity that is only one of the three layers protections. You are right. There are still some cases which CBACs has poor control effect on diseases caused by copper resistant strains. But I need to say, it is also not mean that the induced plant immunity is not important or functional. In some literatures, in order to investigate the treatment of plant disease caused by copper resistant strains, plants have been firstly inoculated with copper resistant strains and then treated with CBACs. This will greatly reduce the effectiveness of CBACs, and the specific reasons need to be analyzed more carefully. Also, from the field practice and our study, we have noticed that CBACs is much more hardly to combined utilize with other pesticides, such as silicon and carbon substrate which will be attenuated the induced plant immunity. Thank you very much for your comments again. We have listed the literature on the poor control effect of CBACs on plant disease caused by copper resistant strains, to give a more balanced overview of copper stories.

  1. Response to comment: Supplementary some literature which indicates that CBACs mixed with EBDC compounds improves the efficacy of controlling not only copper resistance strains, but also highly increases the efficacy of the control of copper sensitive strains that are otherwise controlled.

Response: We are very sorry for the negligence in not listing this part of the work, and we have already added it to the manuscript.

  1. Response to comment: provide a reference in line 338.

Response: We are very sorry for the negligence in not list a reference in line 338, and we have already added it to the manuscript.

  1. Response to comment:The Specific mechanisms that Cu-NPs control plant disease need to be detailed description.

Response: Thank you very much for the precious opinion. We have made detailed revisions to this aspect in the manuscript.

  1. Response to comment: A frequent reference to “13 decades” is awkward.

Response: Thank you for the view. We have replace “thirteen decades” with “fourteen decades or over a century” in the revision.

  1. Response to comment: It is very awkward to refer to “high solubility to scarcely dissolvable reagents”.

Response: Thank you for the view. We have replaced “high solubility to scarcely dissolvable reagents” with “from high solubility to slightly soluble reagents”.

  1. Response to comment: “Break the contacts” is very awkward.

Response: Thank you for the view. We have replaced “Break the contacts” with “Break contact” as the proof-editor’s suggestion.

  1. Response to comment:  It was awkward to start a discussion of copper-based pesticides with a discussion of copper homeostasis and copper biochemistry in plants.

Response: Thank you for the view. We have deleted this piece of content, and added some content on the negative impact of plant diseases and pests on the agricultural development.

  1. Response to comment: The mean of “plant synthesized CuO-NPs”.

Response: We apologize for not explaining the meaning of “plant synthesized CuO-NPs”. It means that extracts from some plants such as Portulaca olracea and Pipper nigram can be used as biological sources for Cu-NPs, achieving the goal of being safe, cheap, ecofriendly.

  1. Response to comment: This would be a good place to refer to older literature by Menkisoglu and others which have talked about the chemistry of copper on plants.

Response: We are very sorry for the negligence in not listing this part of the work, and we have already added it to the manuscript.

  1. Response to comment: This sentence seems to be out of context in that copper compounds are normally applied only to the foliage of plants to inhibit foliar pathogens, and I do not see any relevance about how copper would affect plant roots, unless they are talking about excess copper ions in the soil. This needs to be clarified.

Response: We are very sorry for the lack of a clear explanation. You are right, and we have revised in our manuscript.

  1.  Response to comment: It is very awkward to refer to “even spraying” (perhaps they mean uniform coverage of the leaves) and I also have no idea what an "opacifying agent" is.

Response: Thank you for the view. We have revised in our manuscript.

  1. Response to comment: I'm unfamiliar with the mixing of oils and copper-based pesticides for the control of plant disease. There do not seem to be any references that they cite here. This needs to be better documented.

Response: Thank you for the view. In fact, there may not be much studies on updating additives for CBACs, and there is relatively little research on this area. We have provided a Chinese literature for reference.

  1. Response to comment:  I am very surprised at their discussion of the use of organo- silicone surfactants having relatively low surface tension could be used in conjunction with copper based pesticides. I would expect that this strategy would be unacceptable on many plants that are susceptible to copper ions. The lone reference that they give for the use of organo-silicone surfactants and coppers was for use on citrus, and I would not have expected very much penetration of the leaf with the organo-silicon surfactnts on this species because of the thick cuticle of a typical citrus plant, so I think it may be dangerous to be extrapolating the efficacy of this strategy to other crop species. In fact, on citrus itself, the better dispersal of the applied copper compounds to the very waxy leaves may well have been the main mechanism associated with the increased efficacy when combined with the surfactant.

Response: Thank you for the comments, we have added some explanations in the manuscript that organic silicon additives help CBACs to control citrus canker. Indeed, organic silicon additives are generally used for the prevention and control of citrus diseases. Perhaps more research on surfactants needed to be updated, and we have discussed in “Summary and Future Prospects”.

  1. Response to comment: I would have thought that such a sentence would be one of the very first ones used in the review, rather than buried here most of the way through the manuscript.

Response: Thank you for the view. We have revised to this in the manuscript.

  1. Response to comment: In the first couple of sentences they talk about the effects of copper on pathogenic microbes, but on lines 197, they talk about plant responses. This paragraph needs to be reorganized.

Response: Thank you for the view. We have revised to this in the manuscript.

  1. Response to comment:I think the authors have focused too extensively on a single mechanism of resistance, and I don't think that exclusion of copper is the main mechanism of the resistance in many copper resistant bacteria, and in fact, it was my impression that the Pt23 system that they focus a lot on, actually confers resistance primarily by the sequestration of copper after entry.

Response: Thank you for the view. We have revised to this in the manuscript.

  1. Response to comment: Since I really don't believe in there are three tiered mechanism or resistance, I think Figure 2 needs to be altered to reflect the antimicrobial activity which seems to be main process operating before pathogens enter the plant.

Response: Thank you for your comment. Although sterilization is currently the main viewpoint of CBACs for plant disease control. However, as studies on copper induced plant immune responses become increasingly abundant, the specific mechanisms will also be elucidated in the future. Focusing on plant immune mechanisms will provide a theoretical basis for the more efficient and sustained use of CBACs.

  1. Response to comment: As noted above, I think the authors need to give us more of a background on what is known and what is not known about how copper nanoparticles work. That is, hopefully there is more information than it is currently cited about how efficiently they enter plants and in what form they enter plants etc and whether their toxicity is due to copper ions or special features of the small physical particles themselves.

Response: Thank you for the view. In fact, as a protective fungicide, nano copper is designed to prevent plant disease in a micro and efficient way. Due to its small size characteristics of nanomaterials, the specific surface area rapidly increases, which increases the activity of surface atoms, and increases the free active copper ions multiply. The released copper ions can enter pathogen cells and kill it, but can’t enter plant cells, and there is no evidence to suggest that nano copper can kill pathogens in plant cells.

Reviewer 2 Report

Please make sure you have defined all abbreviated words/names (some of them are highlighted in the attached file).

Please address the comments below and in the attached file.

Comments:

 Lines: 73-77: If only copper ions are toxic to plants, why did they reduce the amount of lime? Please explain.

Line 94: Applicable and scalable method for what? Did they prove that using plant extracts?  And what do you mean by “special plant extracts’?

Line 114: Does it allow the invasion of bacteria only?  Or generally pathogens entry?

Lines: 117 to 119: Do you mean that excess Cu2+ in fungicides can contaminate soil which can cause these adverse effects on plants? Please explain

Lines:c124- 128: Are you talking about one type of BCAs here?  Slightly soluble BCA’s?

Lines: 137 -141: Mineral oil itself added to fungicide spray mixtures is a frequent strategy used to control citrus black spots and potato pests [45, 46]. Is this an example?

If so, I suggest you change the wording as,

The addition/ mixing of mineral oil with fungicide spray could significantly improve the prevention effect of CBACs by enhancing the diffusion, adhesion, and retention of copper ions, increasing the deposition amount of its effective components, and the ability to resist rain wash after being mixed with CBACs. This strategy is frequently used to control citrus black spot and potato pests [45, 46].

Lines: 186-189- First you talk about two systems in bacteria and then three strategies used by pathogenic microbes. Does ‘pathogenic microbes’ mean fungi and bacteria here?  

Lines: 188-189- During the first mechanism, do copper ions pump out of the cell actively using energy stored in ATP? If so tell that as then it would be easy for the reader to understand. And, in the figure, you have CocoG, which you do not talk about in the text.

Line 210-  Figure 1. Two strategies to manipulate the excessive copper in gram-negative bacteria. Are there only two strategies or three strategies?  You explain three in the text. Delete the gap between the figure caption and (a)

Line 241- 243: “To summarize the copper resistance mechanism of Pst, the cop operon, most importantly for CopA andCopB, located on plasmid pPT23D can prevent free copper ions from entering the cytoplasm’.

Suggestion: To summarize the copper resistance mechanism of Pst, the cop operon located  on plasmid pPT23D can prevent free copper ions from entering the cytoplasm through a group of proteins in particular CopA andCopB

Lines 258-261: What are SIMKK and SIMK and what are their functions of these?

Lines 262-263: The idea of this sentence is not clear to me

‘Additionally, the activation of MAPK2, MAPK3, and MAPK4 mediated by excessive 262 copper was found in rice root, which is dependent on ROS’

Lines 272- 273 ‘In Arabidopsis, overexpression miR398 reduces resistance to Pst  DC3000 by inhibiting the transcription of the copper-binding protein, CSD1, and CSD2 .

If this is true, it reduces the resistance. In other words, it increases susceptibility. So, can’t consider a resistant mechanism.

Lines 277-279:  If you are talking about one disease, please name the pathogen.

Lines 296-298: How does this enhance the resistance?

Lines 300-3001: What are StEIN3, StNCED, and StABA1 and how do they enhance the resistance?

Lines 315-316: ‘As a metal ion, copper is not only an essential micronutrient for all organisms, but also the main component of commercial CBACs’. This sentence does not make sense to me. Please reword.

Lines 338-339: ‘For example, commercial Thiodiazole-copper can recover the inhibition activity to pathogens by supplementing Thiodiazole as well as reduce the usage of ionic copper.’ The idea is not clear to me. Do you mean supplementing Thiodiazole-copper with CBAs? 

Author Response

  1. Response to comment: If only copper ions are toxic to plants, why did they reduce the amount of lime? Please explain.

Response: Thank you for your comment. This is our negligence. The hydrated lime relative amount was increased. We have revised to this in the manuscript.

  1. Response to comment: Applicable and scalable method for what? Did they prove that using plant extracts?  And what do you mean by “special plant extracts’?

Response: We apologize for not explaining the meaning of “plant synthesized CuO-NPs”. It means that extracts from some plants such as Portulaca olracea and Pipper nigram can be used as biological sources for Cu-NPs, which is applicable and scalable, so as to achieve the goal of being safe, cheap, ecofriendly. We have revised to this in the manuscript.

  1. Response to comment: Does it allow the invasion of bacteria only?  Or generally pathogens entry.

Response: Thank you for your question. This is our negligence. Indeed, we means it allow the invasion of bacteria. We have revised to this in the manuscript.

  1. Response to comment:Do you mean that excess Cu2+in fungicides can contaminate soil which can cause these adverse effects on plants? Please explain.

Response: We are very sorry for the lack of a clear explanation. We are talking about excess copper ions in the soil could influence plant roots. We have revised to this in the manuscript.

  1. Response to comment: Are you talking about one type of BCAs here?  Slightly soluble BCA’s?

Response: Thank you for your question. In this sentence, we want to express traditional fungicides discussed previously such as Bordeaux mixture.

  1. Response to comment: Mineral oil itself added to fungicide spray mixtures is a frequent strategy used to control citrus black spots and potato pests [45, 46]. Is this an example?

Response: We are very sorry for the lack of a clear explanation. This is an example that talk about the Mineral oil itself added to other fungicide, not CBACs, could control citrus black spots and potato pests. In fact, there may not be much studies on updating additives for CBACs, and there is relatively little research on this area. We have provided a literature about Mineral oil mixed with CBACs could efficiently control plant disease.

  1. Response to comment: First you talk about two systems in bacteria and then three strategies used by pathogenic microbes. Does ‘pathogenic microbes’ mean fungi and bacteria here?  

Response: Thank you for the view. We have revised to this in the manuscript.

  1. Response to comment: During the first mechanism, do copper ions pump out of the cell actively using energy stored in ATP? If so tell that as then it would be easy for the reader to understand. And, in the figure, you have CocoG, which you do not talk about in the text.

Response: Thanks for your questions. Indeed, the incoming Cu2+ is assembled to the cysteine conservative motif of CcoG and con-verted into Cu+ by transferring electrons to the [4Fe-4S] cluster. We have talk about this in line 193-196.

  1. Response to comment: Two strategies to manipulate the excessive copper in gram-negative bacteria. Are there only two strategies or three strategies?  You explain three in the text. Delete the gap between the figure caption and (a).

Response: Thanks for your questions. Not explaining clearly was our negligence. Indeed, two strategies means (a) and (b) in figure 1. three strategies means that the bacteria to maintain the cytoplasmic copper concentration.

  1. Response to comment: Suggestion: To summarize the copper resistance mechanism of Pst, the cop operon located on plasmid pPT23D can prevent free copper ions from entering the cytoplasm through a group of proteins in particular CopA and CopB.

Response: Thank you very much for your suggestion. We have revised this in our manuscript.

  1. Response to comment: What are SIMKK and SIMK and what are their functions of these?

Response: Thanks for your questions. Not explaining clearly was our negligence. SIMK is the mitogen-activated protein kinase (MAPK) of alfalfa (Medicago sativa). SIMKK is the MAPK kinase. We have revised this in our manuscript.

  1. Response to comment: “Additionally, the activation of MAPK2, MAPK3, and MAPK4 mediated by excessive copper was found in rice root, which is dependent on ROS” is unclear.

Response: Thanks for your questions. Not explaining clearly was our negligence. We have revised this in our manuscript.

  1. Response to comment: ‘In Arabidopsis, overexpression miR398 reduces resistance to PstDC3000 by inhibiting the transcription of the copper-binding protein, CSD1, and CSD2 .

If this is true, it reduces the resistance. In other words, it increases susceptibility. So, can’t consider a resistant mechanism.

Response: Thank you very much for the precious opinion. We have revised this in our manuscript.

  1. Response to comment: If you are talking about one disease, please name the pathogen.

Response: Thank you very much for the precious opinion. We have revised this in our manuscript.

  1. Response to comment: Lines 296-298: How does this enhance the resistance?

Response: Thanks for your questions. Not explaining clearly was our negligence. We have revised this in our manuscript.

  1. Response to comment: What are StEIN3, StNCED, and StABA1 and how do they enhance the resistance?

Response: Thanks for your questions. Not explaining clearly was our negligence. StEIN3 is ethylene insensitive 3 of potato (Solanum tuberosum), StNCED is 9-cis-epoxycarotenoid dioxygenase of potato (Solanum tuberosum), StABA1 is ABA biosynthetic genes of potato (Solanum tuberosum). Copper ion (Cu2+) acts as an extremely sensitive elicitor to induce ethylene (ET)-dependent immunity. We have revised this in our manuscript.

  1. Response to comment: ‘As a metal ion, copper is not only an essential micronutrient for all organisms, but also the main component of commercial CBACs’. This sentence does not make sense to me. Please reword.

Response: Thank you very much for the precious opinion. We have revised this in our manuscript.

  1. Response to comment: ‘For example, commercial Thiodiazole-copper can recover the inhibition activity to pathogens by supplementing Thiodiazole as well as reduce the usage of ionic copper.’ The idea is not clear to me. Do you mean supplementing Thiodiazole-copper with CBAs?

Response: Thanks for your questions. Not explaining clearly was our negligence. This means mix thiodiazole with copper to product commercial Thiodiazole-copper. We have revised this in our manuscript.

Round 2

Reviewer 1 Report

The English grammar of this manuscript has been greatly improved. Apparently they had an English editorial improvement company take a look at it, and the manuscript is much easier to read. The authors have also added quite a few new references and discuss some important aspects of the use of copper compounds for plant disease control that had originally not been included, including the use of EBDC fungicides in combination with copper compounds to overcome the resistance of some bacteria to copper. That said, there is a very extensive literature on copper resistant bacteria that is only briefly addressed here. There are two aspects of the manuscript that I had serious issues with in my initial review that have received only minor attention by the authors. The author seem to be completely fixated on their concept of a three-tiered mechanism of action of the copper compounds. While they have made some changes and kind of dance around how the physical barrier made by copper compounds is different from simply the zone of released copper ions, I am still not convinced that this third layer of activity is real or meaningful. Basically, I think it's still boils down to the accumulation of copper ions to a sufficiently i concentration in the surface layer of plants that bacteria or fungi  that are not otherwise resistant to the copper could be killed. The second layer would be their hypothetical induction of resistance mechanisms in the plant by the copper ions. They still emphasize this induced resistance extensively, although there is not very much data on this - mostly by this group in their self citations. I personally think this mechanisms is being over emphasized, although they note that there is more work needed to address this issue. I think they have underemphasize the very extensive literature with that suggest that copper compounds are not effective in controlling copper-resistant bacteria. There is very extensive literature that suggests that prophylactic application of these copper compounds are not effective in controlling copper resistant bacteria. The authors discount the issue of poor control of copper-resistant bacteria by suggesting that the compounds were applied at inappropriate times after infection had already occurred etc. This is clearly not always the case. In fact, I still maintain that the many examples of lack of control of copper resistant bacteria on plants by copper compound is strong evidence that induced disease resistance could not be common or of great magnitude. Therefore, while the manuscript is greatly improved over the original submission, I think it still shows a strong bias that is inappropriate and that some additional changes are needed to put the results of any industry resistance and better context and to once again clarify or justify why we should consider a three-tiered versus a two-tiered mechanism of action of copper compounds.

Author Response

The English grammar of this manuscript has been greatly improved. Apparently they had an English editorial improvement company take a look at it, and the manuscript is much easier to read. The authors have also added quite a few new references and discuss some important aspects of the use of copper compounds for plant disease control that had originally not been included, including the use of EBDC fungicides in combination with copper compounds to overcome the resistance of some bacteria to copper. That said, there is a very extensive literature on copper resistant bacteria that is only briefly addressed here.

Response: Thank you very much for your positive opinion on our polished revision version.

There are two aspects of the manuscript that I had serious issues with in my initial review that have received only minor attention by the authors. The author seem to be completely fixated on their concept of a three-tiered mechanism of action of the copper compounds. While they have made some changes and kind of dance around how the physical barrier made by copper compounds is different from simply the zone of released copper ions, I am still not convinced that this third layer of activity is real or meaningful. Basically, I think it's still boils down to the accumulation of copper ions to a sufficiently i concentration in the surface layer of plants that bacteria or fungi  that are not otherwise resistant to the copper could be killed. The second layer would be their hypothetical induction of resistance mechanisms in the plant by the copper ions. They still emphasize this induced resistance extensively, although there is not very much data on this - mostly by this group in their self citations. I personally think this mechanisms is being over emphasized, although they note that there is more work needed to address this issue. I think they have underemphasize the very extensive literature with that suggest that copper compounds are not effective in controlling copper-resistant bacteria. There is very extensive literature that suggests that prophylactic application of these copper compounds are not effective in controlling copper resistant bacteria. The authors discount the issue of poor control of copper-resistant bacteria by suggesting that the compounds were applied at inappropriate times after infection had already occurred etc. This is clearly not always the case. In fact, I still maintain that the many examples of lack of control of copper resistant bacteria on plants by copper compound is strong evidence that induced disease resistance could not be common or of great magnitude. Therefore, while the manuscript is greatly improved over the original submission, I think it still shows a strong bias that is inappropriate and that some additional changes are needed to put the results of any industry resistance and better context and to once again clarify or justify why we should consider a three-tiered versus a two-tiered mechanism of action of copper compounds.

Response: Thank you very much for your valuable opinions. We won’t emphasize which layer of protection is more importance. Indeed, we think that all three layers of protection are integrated as a spatial system to provide more comprehensive protection. And we think that the two or three-tiered of protections for CBACs is original in conception and be interested to those audience of plant disease control and microbe-plant interactions. It is the novelty and good summarization for this review article. Although we have developed a lot of novel chemicals to control plant disease. But the traditional chemicals, such as CBACs, sulphur powder and lime, are still broadly used for their cost less and familiar operation. We hope to be used for longer term effectiveness of prevention.

Fisrt, we need further explain the induced resistance by copper. In fact, copper induced plant immune response is firstly reported from my group published on Mol Plant (2015), the spraying of CuSO4 or CuCl2 could elicit series of PTI-like responses. Then the copper-induced immune response have been validated to Phytophthora infestans in potato, Rice stripe virus in rice, and Fusarium solani in cucumber, which indicated that copper induced defense in planta is effective to control bacterial, fungal and viral diseases. These findings are recently published from different groups from the year of 2020 to 2022. In annimal system, recently published cuproptosis is also belongs to copper induced immune response which support the conclusion that copper can really induce the plant immunity. 

Second, for your concerns about “many examples of lack of control of copper resistant bacteria on plants by copper compound is strong evidence that induced disease resistance could not be common or of great magnitude.” CBACs have been used as broad-spectrum fungicides. Although many copper-resistant strains have been reported, CBACs are still useful in control pathogens which included copper-resistant strains. We have discussed this in lines 334-337 and 362-364 of the manuscript. For those lack of control of resistant bacteria, I cannot get a definite answer from the existing literature. In fact, the defense signaling pathway induced by copper is a PTI-like response, such PTI could be defeated by so many pathogen and effector-triggered susceptibility that means not to control all plant diseases. However, we can’t agree that PTI has no useful to control disease.We have also explained this in line 402-405. Once again, for lack of control, we didn’t clear what is really happened for copper-tolerant strains except copper homeostasis and cop resistant system. We had uncovered that pathogenic Xanthomonas oryzae PXO99 is more sensitive to copper caused by copA mutation (Yuan et al. Plant Cell, 2010; Kong et al., Acta Phytopathologica Sinica, 2018, in Chinese). To overcome the copper sensitive, it seems not to develop another copper-tolerance strategies but to a novel TAL effector of PthXo1 to upregulate the expression of plant copper-uptake complex of OsSWEET 11/Xa13-COPT1-COPT5 to reduce the copper concentration in vascular bundle (Yang et al. PNAS, 2006). Moreover, the upregulated OsSWEET11/Xa13 protein has additional susceptible functions as the sucrose efflux from the phloem parenchyma cells to the apoplast for bacterial proliferation (Chen et al., Nature, 2010). We have supplemented the talk in the line 312-321.

Regarding the proposed three-layer protection theory, many researchers believe that the slightly soluble CBACs cover the surfaces of plant is to better exert their bactericidal effects, which seems to merge the first and second layers. Therefore, we have previously made modifications to the title of this article. However, in this manuscript, we would like to emphasize that CBACs can form a breathable, light permeable, and water permeable membrane on the surface of plants, which can prevent direct contact between pathogens and plants. As we known, physical separation such as paper bag for fruits are very popular and welcome used in orchard. And we have made further modifications in our manuscript, which is highlighted in blue in newest manuscript. At present, more and more researchers are aware of such a role and use it to better develop Cu-NP materials, and prove the physical barrier through SEM, promote the better development of CBACs. Thank you again for your valuable feedback.